# Climatic controls on mountain glacier basal thermal regimes dictate spatial patterns of glacial erosion

Jingtao Lai[1,2], Alison M. Anders[1]

[1]Department of Geology, University of Illinois at Urbana-Champaign, Urbana, IL, 61801, USA
5    [2]GFZ German Research Centre for Geosciences, Potsdam, 14473, Germany

*Correspondence to*: Jingtao Lai (lai@gfz-potsdam.de)

For submission to *Earth Surface Dynamics*

**Abstract**

10    Climate has been viewed as a primary control on the rates and patterns of glacial erosion, yet our understanding of the mechanisms by which climate influences glacial erosion is limited. We hypothesize that climate controls the patterns of glacial erosion by altering the basal thermal regime of glaciers. The basal thermal regime is a first-order control on the spatial patterns of glacial erosion. Polythermal glaciers contain both cold-based portions that protect bedrock from erosion and warm-based portions that actively erode bedrock. In this study, we model the impact of various climatic conditions on glacier basal thermal regimes and patterns of glacial erosion in mountainous regions. We couple a sliding-dependent glacial erosion model with the Parallel Ice Sheet Model (PISM) to simulate the evolution of the glacier basal thermal regime and glacial erosion in a synthetic landscape. We find that both basal thermal regimes and glacial erosion patterns are sensitive to climatic conditions and glacial erosion patterns follow the patterns of the basal thermal regime. Cold temperature leads to limited glacial erosion at high elevations due to cold-based conditions. Increasing precipitation can overcome the impact of cold temperature on the basal thermal regime by accumulating thick ice and lowering the melting point of ice at the base of glaciers. High precipitation rates, therefore, tend to cause warm-based conditions at high elevations, resulting in intensive erosion near the peak of the mountain range. Previous studies often assess the impact of climate on the spatial patterns of glacial erosion by integrating climatic conditions into the equilibrium line altitudes (ELAs) of glaciers, and glacial erosion is suggested to be maximal around the ELA. However, our results show that different climatic conditions produce glaciers with similar ELAs but different patterns of basal thermal regime and glacial erosion, suggesting that there might not be any direct correlation between ELAs and glacial erosion patterns.

## 1 Introduction

Earth's past climate has left a clear imprint on the topography of mountain ranges worldwide. During the late Cenozoic, global cooling induced widespread glaciation and glacial erosion created unique landforms in mountainous regions, such as cirques, hanging valleys, and overdeepenings. Climate is a primary control on the pace and spatial variability of glacial erosion and better constraint on this control is essential to improve understanding of the development of topography worldwide during the climate perturbations of the late Cenozoic. Recent compilations of modern glacial erosion rates have provided an empirical measure of the relationship between climate and glacial erosion (Cook et al., 2020; Koppes et al., 2015). Temporal evolution of glacial erosion rates inferred from sedimentary records also suggest that glacial erosion mostly occurs in some optimal climatic conditions (Fernandez et al., 2011; Ganti et al., 2016; Mariotti et al., 2021). Yet a process-based understanding of how climatic conditions influence the rates and patterns of glacial erosion is still limited. Intuitively, climate could influence glacial erosion by modulating the thermal structures of glaciers, because warm-based glaciers are much more powerful erosional agents than cold-based glaciers (Kleman and Glasser, 2007). We explore this idea by using numerical simulations to investigate the impact of climatic conditions on the basal thermal regime of glaciers and, consequently, the rates and patterns of glacial erosion.

Climatic controls on glacial erosion have often been assessed by integrating climatic conditions into the equilibrium line altitudes (ELAs) of glaciers. Previous studies have suggested that glacial erosion is most effective at or above the ELA of a glacier (e.g., Anderson et al., 2006; MacGregor et al., 2000). Numerical landscape evolution models that approximate the erosion rate as a function of sliding velocity also produce focused erosion near the ELA (e.g., Herman et al., 2011; MacGregor et al., 2000). In addition, the strong correlation between the mean or peak elevation of mountains and the ELAs of modern or past glaciers in some mid-latitude mountain ranges suggests that glacial erosion is concentrated near or above the ELA (Anders et al., 2010; Brozović et al., 1997; Egholm et al., 2009; Mitchell and Montgomery, 2006). However, the correlation between the ELAs and mountain heights breaks down in high-latitude mountain ranges because the cold-based glaciers at high elevations cause limited erosion, resulting in high mountain peaks that sit above the ELA (Thomson et al., 2010). Additionally, measurements of sediment production by modern glaciers reveal that the rates of glacial erosion vary as a function of latitude, which is an indication of the basal thermal regime (Koppes et al., 2015). These observations suggest that the basal thermal regime is a fundamental control on the rates and spatial patterns of glacial erosion and motivate us to consider the influence of climate on the basal thermal regime, rather than the ELA, as a primary control on glacial erosion.

The basal thermal regime is expected to exert first-order control on the spatial variability in glacial erosion. Basal sliding speed and meltwater pressure both strongly modulate the rate of glacial abrasion and quarrying (Hallet, 1979, 1996; Iverson, 2012) and are both controlled by the basal thermal regime. Below cold-based glaciers, the basal ice is frozen to the bedrock and limited basal sliding and meltwater supply cause minimal glacial erosion. In contrast, warm-based glaciers erode their beds via abrasion and quarrying due to active basal sliding and meltwater production. Under large continental ice sheets, the contrast in erosive power between cold-based and warm-based portions of the ice sheets has been suggested to have caused

selective linear erosion of deep valleys and fjords along glaciated continental margins (Hall et al., 2013; Kleman and Glasser, 2007).

While polythermal glaciers that contain both warm-based and cold-based portions are common in mountainous regions, the influence of the basal thermal regime on the erosion of polythermal alpine glaciers has received little study. Previous glacial landscape evolution models often neglect the basal thermal regime by assuming the glacier is entirely warm-based (e.g.,

MacGregor et al., 2000; Prasicek et al., 2018). A few studies have examined polythermal mountain glaciers and demonstrated that a cold climate may produce cold-based ice at high elevations (Anderson et al., 2012; Tomkin and Braun, 2002; Yanites and Ehlers, 2012). However, the glacier thermodynamics in these early glacial landscape evolution models is oversimplified. The basal temperature is approximated by using a one-dimensional column model that accounts for the vertical heat transportation and neglects the longitudinal component (e.g., Tomkin and Braun, 2002). Recent compilation of modern glacial

erosion rates highlights the complex relationships between climate, glacier dynamics, and glacial erosion (Cook et al., 2020; Koppes et al., 2015). Therefore, a better approximation for the glacier thermodynamics is essential in glacial landscape evolution modeling. In our previous work (Lai and Anders, 2020), we built a landscape evolution model that includes a more sophisticated representation of thermodynamics (Aschwanden et al., 2012). Our previous focus was on how geothermal heat fluxes influence the basal thermal regime and glacial erosion. In this study, we use our glacial landscape evolution model with

a thermodynamically coupled ice dynamics model to investigate the climatic control on the rates and patterns of glacial erosion through the basal thermal regime. We aim to explore the influence of precipitation and temperature on the spatial pattern of glacial erosion that arises through modulation of the basal thermal regime. We present a series of numerical simulations that allow us to assess the correlation between the basal thermal regimes of glaciers and the rates and patterns of sliding-driven glacial erosion under a range of climatic settings.

**2 Methods**

We build a landscape evolution model with the Parallel Ice Sheet Model (PISM, http://www.pism-docs.org) to simulate the evolution of glacial landscapes. The approach we use in this study is similar to that presented in Lai and Anders (2020) where we first added glacial erosion to PISM. In this study, we extend the model presented by Lai and Anders (2020) by adding fluvial incision and bedrock uplift to the landscape evolution model. In this section, we briefly summarize the different

components of our model.

**2.1 Ice flow model – Parallel Ice Sheet Model**

To solve for ice flow, PISM uses a hybrid stress balance scheme that combines the Shallow Ice Approximation (SIA; Hutter, 1983) for internal deformation and the Shallow Shelf Approximation (SSA; Morland, 1987) for membrane stress (also known as longitudinal stress). The membrane stress is an important component in balancing the driving stress in alpine glaciers

(Bueler and Brown, 2009; Hindmarsh, 2006). Basal sliding velocity is related to the basal shear stress through a Weertman-

style sliding rule and it is controlled by the balance between basal shear stress, membrane stress, and driving stress. Basal sliding velocity is also controlled by the amount of subglacial meltwater through a simple subglacial hydrology model. The conservation of energy is solved using an enthalpy-based scheme in PISM (Aschwanden et al., 2012). The governing equations of PISM are presented in Bueler and Brown (2009) and Winkelmann et al. (2011) and we refer readers to these works for a detailed description of the model.

PISM has been used to simulate the contemporary Greenland Ice Sheet and the result shows a good correlation between modeled and observed ice surface velocity (Aschwanden et al., 2016). PISM has also been used to reconstruct the complex history of glaciation in mountainous regions (e.g., Golledge et al., 2012; Seguinot et al., 2018).

## 2.2 Landscape evolution model

The evolution of bedrock topography is controlled by glacial erosion, fluvial incision, and uplift. At each time step, bedrock topography is uplifted at a uniform and constant rate across the model domain. In areas where the thickness of ice is greater than 10 m, only glacial erosion can change the topography, and in other areas, only fluvial incision is allowed to occur. We assume that all eroded materials are transported out of the model domain efficiently so that there is no deposition in the system.

### 2.2.1 Glacial erosion model

The rate of glacial erosion, $E_g$, is modeled as a linear function of the sliding velocity, $\boldsymbol{u}_s$:

$$E_g = K_g |\boldsymbol{u}_s| , \tag{1}$$

where $K_g$ is an erodibility coefficient. In this study, the value of $K_g$ is 0.0001 in all simulations. This erosion model has been widely used in glacial landscape evolution models (e.g., Egholm et al., 2011; Herman et al., 2011; MacGregor et al., 2000; Tomkin and Braun, 2002; Yanites and Ehlers, 2012). Although available field measurements have suggested a nonlinear relationship between sliding velocity and glacial erosion rates (Cook et al., 2020; Herman et al., 2015; Koppes et al., 2015), we choose to use a linear erosion rule for simplicity, since our main goal is to investigate the influence of climate on the spatial patterns of erosion. This model is supported by theoretical studies of glacial abrasion (Hallet, 1979) and it is a reasonable approximation of glacial erosion when abrasion dominates glacial erosion (Humphrey and Raymond, 1994). Although glacial erosion by quarrying is complicated by the subglacial hydrological conditions (Hallet, 1996; Iverson, 2012), this sliding-dependent model still reproduces the qualitative patterns of glacial erosion from a numerical model driven by a quarrying law (Ugelvig et al., 2016). A common shortcoming of this sliding based model is that steep bedrock slopes can produce unrealistically high erosion rates and trigger runaway effects (Herman et al., 2011). To avoid this, we do not allow bedrock slopes to exceed a threshold value of 45°. If the slope of bedrock topography reaches the threshold value, glacial erosion is prohibited.

### 2.2.2 Fluvial incision model

Fluvial incision is modeled using the stream power incision model (Whipple and Tucker, 1999). The rate of fluvial incision, $E_f$, is a function of drainage area, $A$, and bedrock slope, $S$:

$$E_f = K_f A^m S^n, \tag{2}$$

where $K_f$ is an erodibility coefficient and $m$ and $n$ are constants. $K_f$ is a major unconstrained parameter in the stream power incision model (Harel et al., 2016). In this study, we choose to use a value of 0.00001 because it falls within the typical range of $K_f$ used in many previous studies (e.g., Herman and Braun, 2008; Lai and Anders, 2018; Whipple and Tucker, 1999) and it predicts a reasonable fluvial relief in our simulations. The value of $K_f$ is uniform across the model domain and constant over the glacial-interglacial cycle. The $m/n$ ratio is predicted to be ~0.5 by theory (Whipple and Tucker, 1999) and it is supported

by global field observations (Harel et al., 2016). In our simulations, $m$ and $n$ are 0.5 and 1, respectively. Flow direction is approximated using the D8 algorithm and the drainage area is calculated using the Fastscape algorithm (Braun and Willett, 2013). In our implementation, the drainage area includes upstream areas occupied by glaciers. In glaciated areas, the direction of water flow is determined based on ice surface elevation rather than bedrock elevation. Fluvial incision only applies to the areas outside of the glacial realm, and in glaciated areas, the rate of fluvial incision is set to zero.

Ideally, the fluvial incision model should reflect the influence of glacier meltwater and precipitation on fluvial incision. However, the goal of this study is to investigate the climatic controls on glacial erosion through the basal thermal regime, and incorporating a climate-dependent fluvial incision model could make it difficult to isolate the impact of climatic conditions on glacial erosion. Therefore, we simply model fluvial incision using the stream power incision law.

### 2.3 Initial conditions

The initial bedrock topography is a synthetic fluvial landscape created in the Landlab model platform (Hobley et al., 2017). The fluvial landscape is a 100-km by 100-km mountain range with 20-km wide piedmont plains on each side (Fig. 1a). The piedmont plains are removed in all figures for a clear illustration of the mountain range. The fluvial incision model used for creating the initial topography is the same as the model described in Section 2.2. and the value of the fluvial erodibility coefficient is also 0.00001. The rate of uplift is 0.0035 m year[-1]. The uplift rate and fluvial erodibility coefficient used for

creating the initial topography are maintained in the subsequent glacial erosion simulations. Fluvial incision and rock uplift are in equilibrium in the initial topography such that the fluvial incision rate equals the rock uplift rate. The initial topography has a relief of ~3000 m and the mountain range has 5 major valleys on each side. The grid resolution is 1 km. This resolution is chosen because it provides a reasonable balance between accuracy and efficiency in PISM (Aschwanden et al., 2016).

All the simulations start from an ice-free topography. This is a reasonable initial state because in most cases the climate

forcing only allows for a limited ice cover along the mountain ridges during the interglacial periods.

## 2.4 Climate forcing

Climate forcing is represented by the mean annual sea-level temperature and mean annual precipitation, and PISM takes these two parameters as input values to calculate the ice surface mass balance. Spatially, the mean annual temperature decreases as the elevation rises with a lapse rate of 6.5 °C km$^{-1}$, and the mean annual precipitation is uniform across the model domain. Temporally, the seasonal variation of temperature is modeled by a sinusoidal function with the summer temperature assumed to be 5 °C higher than the mean annual temperature. There is no seasonal variation in precipitation. A positive degree day (PDD; Calov and Greve, 2005) model then calculates the ice surface mass balance based on temperature and precipitation.

In all simulations, we use a 100,000-year glacial-interglacial cycle with a "saw-tooth" variation of temperature (Fig. 1b). The mean annual sea-level temperature decreases by 8 °C linearly for 80,000 years and then increases linearly for 20,000 years. The mean annual precipitation increases by 7.2% for every one degree Celsius of increase in temperature (Huybrechts, 2002).

## 2.5 Experiment design

We explore the impact of climatic conditions on glacial erosion by varying the mean annual sea-level temperature and mean annual precipitation at the glacial maximum. The glacial mean annual sea-level temperature ranges from 1 to 5 °C and the mean annual precipitation at glacial maximum ranges from 50 to 2000 mm year$^{-1}$. In order to isolate the impact of basal thermal regime on glacial erosion from the influence of glacier extent and ELAs, we select different ranges of precipitation rates for different temperature values. For each temperature value, through trial and error, we first choose a proper precipitation rate that allows the glacier fronts to reach the edge of the mountains and then we explore a list of precipitation rates below this value. This allows us to conduct a group of simulations with similar ELAs. However, the range of precipitation for cold climates is small because cold climates produce large glaciers without significant amounts of precipitation. Therefore, we conduct an additional group of simulations with cold temperatures and high precipitation rates. The values of mean annual sea-level temperature and mean annual precipitation for all the simulations are summarized in Table 1.

For each climate condition, we examine not only the output of our landscape evolution model, but also consider the output from PISM over an unchanging topography. These glaciation-only cases isolate the impact of climate on the basal thermal regime because they avoid any feedbacks between evolving topography and the glacier basal thermal regime. All the parameters in the landscape evolution models including the glacial erosion coefficient, the stream power erosion coefficient, and the bedrock uplift rate are held constant in all the simulations. All the simulations are run over one 100,000-year glacial-interglacial cycle.

## 3 Results

In order to highlight the climatic controls on the basal thermal regime of glaciers and spatial patterns of glacial erosion, we first compare a set of models in which different climate conditions produce similar ELAs at the glacial maximum. Next,

we compare the results of groups of simulations with different mean annual sea-level temperatures and the same mean annual precipitation rate at the glacial maximum to explore the sensitivity of the spatial pattern of glacial erosion to temperature. Finally, we compare the results of cases with different mean annual precipitation rates and the same mean annual sea-level

temperature at the glacial maximum to investigate the influence of precipitation.

### 3.1 Climatic controls on the basal thermal regime

We begin by exploring the sensitivity of basal thermal regimes to climatic conditions by comparing results of glaciation-only cases in which landscape evolution models are not enabled. In order to isolate the impact of glacier sizes and ELAs on glacial erosion, we compare the results of three simulations with similar ELAs at the glacial maximum but different climatic

conditions. Unsurprisingly, the basal thermal regimes of simulated glaciers are distinct in each case and strongly controlled by climatic conditions, despite the similarity in the ELA and ice extent across all the cases (Fig. 2). Different climatic conditions in the three simulations produce similar ELAs around 1300m at glacial maximum. As a result, the modeled extent and thickness of ice at the glacial maximum is also similar in different cases (Fig. 2a-c). The basal thermal regimes at glacial maximum, however, vary significantly as a function of climate despite the similar ice extent and thickness (Fig. 2d-f). In a cold and dry

climate (1 °C, 400 mm year$^{-1}$), warm basal ice only occurs in major valleys, while glaciers at high elevations are mostly cold-based due to the cold temperature (Fig. 2d). As the climate transitions into warmer conditions, glaciers near the center of the range shift to warm-based conditions, and areas with warm basal ice extend into higher elevations (Fig. 2e). In the warmest climate (5 °C, 1600 mm year$^{-1}$) most of the glaciers are warm-based (Fig. 2f). The different basal thermal regimes have the potential for producing distinct glacial erosion patterns, as we will show in the next section.

In addition to the basal thermal regime, basal shear stress is another important control on basal sliding, and consequently, glacial erosion (Seguinot and Delaney, 2021). All the three simulations predict high shear stress along mountain ridges and in major valleys, and the spatial patterns of basal shear stress show much less variations between different climates than the patterns of basal thermal regime, especially in major valleys (Fig. 2g-i).

### 3.2 Spatial patterns of erosion controlled by basal thermal regime

Having demonstrated that climate strongly influences the distribution of warm ice in the absence of erosion, we now implement glacial and fluvial erosion and rock uplift to compare the modeled glacial erosion in three cases with different climates but similar ELAs. We quantify the average basal thermal regimes over a glacial-interglacial cycle by calculating the percentage of time with warm-based conditions during a cycle. In all simulations, glacial erosion tends to focus in areas where the basal ice is mostly warm throughout the whole cycle (Figs. 3 and 4). In the case with a cold and dry climate (1 °C, 400 mm

year$^{-1}$), glaciers are perennially cold-based at high elevations (Figs. 3g and 4d), leading to limited glacial erosion at high elevations near the center of the range (Figs. 3d and 4a). Warm-based areas are mostly found in major valleys (Figs. 3g and 4d). During a glacial-interglacial cycle, middle parts of the valleys are influenced by warm-based glaciers for a longer period than lower parts of the valley (Figs. 3g and 4d) because the lower parts are only covered by glacial ice for a limited period

during the coldest intervals. Consequently, most glacial erosion occurs in the middle parts of major valleys (Figs. 3d and 4a).

In contrast, in a warm and wet climate (5 °C, 1600 mm year$^{-1}$), warm-based areas extend into higher elevations than in a cold and dry climate and glaciers are constantly warm-based at high elevations (Figs. 3i and 4f). The area with significant glacial erosion also migrates towards the center of the range at high elevations in a warm and wet climate (Figs. 3f and 4c).

The different spatial patterns of glacial erosion lead to distinct landforms in different climates. In a cold and dry climate, the glacial erosion rate exceeds the bedrock uplift rate in major valleys, producing overdeepenings and increasing local relief, 220 while at high elevations, pre-glacial landforms are preserved under cold-based glaciers and a limited amount of erosion allows for an increase of the elevation of some peaks (Figs. 3a and 5a). In contrast, in a warm and wet climate, significant erosion at high elevations lowers the peaks and efficiently reshapes the topography near the center of the range, creating cirque-like landforms and overdeepenings near the peaks (Figs. 3c and 5c). Distinct landscapes caused by variation in basal thermal regimes are also reflected by changes in the hypsometry of the topography (Fig. 6). In a cold and dry climate, the relief of the 225 mountain range is increased after a glacial-interglacial cycle, while the relief is decreased in a warm and wet climate, even when the ELAs at the glacial maximum are similar.

We have shown that the spatial pattern of erosion accumulated throughout an entire glacial cycle varies due to climatic effects on the basal thermal regime. Additionally, we observe that erosion rates at different stages during a glacial cycle also reveal the influence of climate on glacial erosion patterns through the basal thermal regime (Fig. 7). Early in the glacial cycle, 230 all three climates predict limited ice cover at high-elevations near the center of the mountain range. However, the warm and wet case features much greater erosion rates than the colder and dryer cases (Fig. 7a-b), despite similar extents of ice cover. Similarly, at the glacial maximum, the spatial patterns of erosion are different under different climatic conditions (Fig. 7d-f) even though these climates produce glaciers with similar sizes. In a cold and dry climate, most erosion occurs at low-elevations in major valleys (Fig. 7d), while a warm and wet climate predicts focused erosion at high elevations (Fig. 7f). During the 235 deglaciation stage, the case with warm and wet climate has lower erosion rates than the other two cases (Fig. 7g-i), because the topography is eroded and the size of glaciers is limited.

### 3.3 Sensitivity to temperature

Air temperature is one of the primary controls on the glacier basal thermal regime. We compare cases with different mean annual sea-level temperatures and the same precipitation rate at the glacial maximum. Unsurprisingly, the extent of glaciation 240 is strongly controlled by the air temperature. In a warm climate, glaciers are restricted to the upper part of the mountain range due to the relatively high ELA, while in cooler climates the majority of the mountain range is influenced by glaciation (Fig. 8).

Glaciers in a warm climate are mostly warm-based throughout the cycle and most glacial erosion occurs at high elevations because high elevation regions are influenced by warm basal ice for a longer period than lower elevations (Figs. 8 and 10). As 245 the climate transitions from a warm one into a cold one, it is commonly expected that the basal thermal regime at high elevations will shift from warm-based to cold-based. In our simulations, we observe such transition in basal thermal regime in

relatively dry climates. In a dry and warm climate, the glaciers are mostly warm-based and are restricted within high elevation regions, causing a small amount of glacial erosion primarily focusing on the center of the range (Fig. 8b). As the temperature decreases, glaciers at high elevations transition into cold-based conditions, resulting in limited glacial erosion (Fig. 8a). In contrast, in relatively wet climates, decreases in temperatures do not lead to a transition from warm-based to cold-based conditions (Fig. 8d-f). In a cold but relatively wet climate, high elevation regions are still covered by warm-based rather than cold-based glaciers, allowing for a great amount of erosion at high elevations (Fig. 8d). This indicates that the sensitivities of glacier basal thermal regimes and glacial erosion to air temperature are dependent on the precipitation rates. A relatively wet climate could allow for warm-based areas at high elevations even in a cold climate. In the next section, we will further investigate the influence of precipitation on basal thermal regimes and glacial erosion.

### 3.4 Sensitivity to precipitation

We compare cases with different mean annual precipitation rates but the same air temperature at the glacial maximum. Increasing precipitation lowers the ELA by expanding the accumulation zone of glaciers. As expected, glaciers are smaller in a dry climate than in a wet climate, resulting in less glacial erosion (Figs. 9 and 10). There is a potential for a larger warm-based area in a wet climate than a dry climate because the thick ice in a wet climate lowers the melting point of ice and works to prevent the dissipation of heat accumulated at the base of ice. Increasing precipitation in cold climates allows warm-based ice to occur at increasingly high elevations. As a result, in cold climates, the area with significant erosion migrates into high elevations toward the center of the range as the climate becomes wetter (Figs. 9a-c) despite that the ELAs are lowered by high precipitation rates. In contrast, increasing precipitation in warm climates has little impact on the basal thermal regime because the glaciers are mostly warm-based already. In warm climates, glacial erosion constantly focuses at high elevations as the precipitation increases (Figs. 9d-f), although the glaciers become larger in a wetter climate.

### 3.5 Synthesis - climatic controls on the spatial patterns of glacial erosion via basal thermal regime

We quantify the spatial patterns of glacial erosion by identifying the "median location of erosion". For each simulation, we scan the eroded topography starting from both fronts of the mountain range until the scanned area consists of 50 % of the total amount of glacial erosion. This location is described by the distance from the range fronts and we refer this distance as the "median location of erosion". The median location of erosion is greater (closer to the ridge center) for a simulation predicting that glacial erosion concentrates near the center of the mountain than a case in which glacial erosion focuses near the fronts of the mountain range.

The median location of erosion integrates the spatial distribution of glacial erosion into one single value and allows for a systemic comparison of glacial erosion patterns across the range of climatic scenarios explored in this study. In general, warm climates result in median locations of erosion that are closer to the center of the mountain range than cold climates (Fig. 10a), because warm climates lead to warm-based conditions in high-elevation regions and restrict the distribution of ice near the center of the mountain. The influence of precipitation on the spatial patterns of glacial erosion is also revealed by the median

location of erosion. In warm climates, the median locations of glacial erosion are close to the center of the mountain range and as the precipitation increases, the median locations migrate slightly towards the edge of the mountain range due to increased glacially influenced area (Fig. 10a). A similar trend is also observed in cold climates when the precipitation is low. For example, when the mean annual temperature at sea-level is 1 °C during glacial maximum, the median locations of glacial erosion also migrate toward the edge of the mountain range as precipitation rises up to ~300 mm year$^{-1}$. If the precipitation rates keep increasing, the median locations of glacial erosion will move towards the center of the mountain range because of the increased warm-based conditions at high elevations (Section 3.4; Fig. 10a-c). This finding suggests that the dependency of glacial erosion patterns on precipitation is more complicated in cold climates than warm climates.

We also summarize the ELAs and the fraction of warm-based area in glaciated regions during the glacial maximum for glaciation-only cases (Fig. 10b). The rates of glacial erosion are generally correlated with the fraction of warm-based area in glaciated regions. High fractions of warm-based areas correspond to fast rates of glacial erosion (Fig. 10a and 10b). The spatial patterns of glacial erosion reflected by the median location of glacial erosion do not always follow the patterns of ELAs. In warm climates, the ELAs are lowering as the precipitation rates increases, and both median locations of glacial erosion and the intersections between the ELA and the topography are migrating towards the front of the mountain range. However, in cold climates, as the precipitation rates increase, the median locations of erosion migrate towards the edge of the mountain range first, and then move back towards the center of the mountain range, while the ELAs are lowering constantly and the intersections of ELA and topography migrate towards the edge of the mountain range. The different sensitivities to climates between the spatial patterns of glacial erosion and the ELAs suggest that the spatial patterns of glacial erosion are not fully controlled by the ELAs, especially in cold climates.

## 4 Discussion

### 4.1 ELA, basal thermal regime, and the location of maximum glacial erosion

Previous studies of glacial erosion and glacial landscapes have emphasized the role of ELA in controlling the spatial patterns of erosion. The correlation between ELA and the spatial patterns of erosion partially arises from a simple framework: if we assume the rate of glacial erosion to a first-order scales with ice discharge (Anderson et al., 2006), then glacial erosion tends to focus around the ELA because ice discharge peaks at the ELA. Although ice discharge is a convenient proxy for erosion, many studies have shown that glacial erosion is controlled by sliding velocity (Hallet, 1979; Herman et al., 2015), subglacial hydrology (Beaud et al., 2014; Herman et al., 2011), and basal thermal regime (Koppes et al., 2015). In temperate glaciers with mostly warm basal ice, basal sliding occurs throughout the whole glacier, and therefore, basal sliding velocity scales, to first order, with ice discharge. Subglacial meltwater, however, tends to focus in the ablation zone and promotes sliding and erosion in low elevation areas (Herman et al., 2011). The basal thermal regime is not correlated with ice discharge or ELA. Our previous work (Lai and Anders, 2020) showed that geothermal heat from the underlying bedrock can significantly change the basal thermal regime of glaciers without any changes in surface conditions, including the ELA. In this study, our

numerical simulations show that the trade-off between temperature and precipitation could results in glaciers with similar ELAs but different basal thermal regimes (Fig. 2) as well as distinct patterns of glacial erosion (Figs. 3 and 4). Our results indicate that the patterns of glacial erosion are closely tied with the basal thermal regime rather than the ELA. Overall, based on our results and previous studies, we suggest that there might not be any direct spatial correlation between the ELA and the location of maximum erosion.

The observed agreement between mountain peak elevations and reconstructed past ELAs, i.e., the glacial buzzsaw hypothesis (Brozović et al., 1997; Egholm et al., 2009; Mitchell and Montgomery, 2006), suggests glaciers might focus their erosion at or above the ELAs. However, the past ELAs are often reconstructed using the cirque floor elevations (Mitchell and Montgomery, 2006; Porter, 1989, 2000), and they might represent the average glacial conditions rather than the actual ELA determined by a specific climate (Barr and Spagnolo, 2015; Porter, 1989). Cirques are formed over multiple glacial/interglacial cycles and the development of a cirque is thought to primarily occur during periods with modest climate when the glacier is restricted within the cirque and is mostly warm-based (Barr and Spagnolo, 2015). The cirque floor elevations, therefore, are determined by the average intermediate conditions over multiple glacial/interglacial cycles (Barr and Spagnolo, 2015; Porter, 1989). As the cooling climate leads to more extensive glaciations, cirque enlargement might cease because the cirque is covered by cold-based ice, and the climatic conditions during these more extensive glaciation periods are not recorded in cirques. Our model results show that, although periods with extensive glaciation only occupy a short time interval of the whole glacial/interglacial cycle, the warm-based valley glaciers produce large amounts of erosion in major valleys during periods with extensive glaciation (Fig. 7). This observation from numerical simulations is also supported by the presence of widespread overdeepenings in glaciated mountain ranges (Magrani et al., 2020). For this reason, we suggest that cirque-based ELA estimates might not be an appropriate proxy for assessing the influence of past climate on glacial erosion, and their correlation with mountain peak elevations cannot support the idea that climate controls the spatial patterns of glacial erosion via changing ELAs. Observations of cirque floor elevation and cirque headwall relief suggest that cirques may set the base level for the hillslope processes that potentially limit the mountain peak elevations (Anders et al., 2010; Mitchell and Montgomery, 2006), and therefore, we speculate that the observed trend is the correlation between peak elevations and planes defined by cirque floors.

## 4.2 Implication for understanding the sensitivity of glacial erosion to climate

While precipitation has been viewed as the primary driver of fluvial incision (e.g., Ferrier et al., 2013) and hillslope erosion (e.g., Moon et al., 2011), the role of precipitation in controlling the rates and patterns of glacial erosion has received limited study. In this work, we observe a wide range of glacial erosion rates as a function of precipitation. The rate of glacial erosion increases by 2 orders of magnitude as the precipitation rate rises by a factor of 5-10 (Fig. 10a). In cold conditions, increases in precipitation could change the basal thermal regime and cause a large amount of erosion at high elevations (Fig. 9a-c). Most previous studies focusing on the impact of climate on glacial erosion have put an emphasis on the role of temperature in lowering the ELAs and in controlling basal thermal regime (e.g., Thomson et al., 2010; Yanites and Ehlers, 2012). It is often

suggested that glacial erosion is lower in cold, high-latitude regions because the cold temperature implies more frequent cold-
based conditions. However, our simulations show that high precipitation rates could overcome the influence of cold
temperature on the basal thermal regime by accumulating thick ice and lowering the melting point of ice. Precipitation also
has the potential to promote basal sliding, glacial erosion, and the evacuation of sediments if the liquid water is able to reach
the bed of glaciers (Cook et al., 2020; Herman et al., 2011; Koppes et al., 2015). The correlation between Quaternary erosion
hotspots and precipitation maxima in the Patagonian Andes also suggests that precipitation exerts a first-order control on
glacial erosion rates (Herman and Brandon, 2015). A recent global compilation of modern glacial erosion rates even suggests
that precipitation explains more of the variability of modern erosion rates than temperature (Cook et al., 2020). Therefore, we
suggest that precipitation should be viewed as equally important as temperature when assessing the influence of climate on
glacial erosion.

Our simulations also show that increasing precipitation could result in a drop in the ELA, and this finding is consistent
with field observations (Oien et al., 2020). However, in most previous glacial landscape evolution models, precipitation is
often integrated into the mass balance term or changes as a function of temperature, and the impact of precipitation on the ELA
is not explicitly modeled (e.g., Yanites and Ehlers, 2012). We suggest that precipitation should be viewed as an independent
component in glacial landscape evolution models.

Koppes et al. (2015) observed significant latitudinal variation of contemporary glacial erosion rates in the Patagonia and
Antarctic Peninsula and they suggested that a mean annual temperature around 0-5 °C might represent a threshold condition
for fast glacial erosion due to shifts between cold-based to warm-based conditions. In our work, the explored range of the mean
annual sea-level temperatures lies in this threshold range, but our results do not show a significant increase in glacial erosion
rates as mean annual temperatures increase from 1 to 5 °C (Fig. 10a). Instead, our numerical simulations predict that climates
with low temperatures and high precipitation rates are optimal conditions for glacial erosion (Fig. 10a). A recent global
compilation of modern erosion rates also shows a pattern of high glacial erosion rates in similar climatic conditions (Cook et
al., 2020). This is probably because the glaciers surveyed by Koppes et al. (2015) are all large outlet tidewater glaciers with
similar catchment sizes, while in our simulations, the sizes of glaciers are highly variable. For example, in a warm and dry
climate, the sizes of ice bodies are not large enough to form fast-flowing valley glaciers, and as a result, the rates of glacial
erosion are limited (Fig. 8c; the lower right corner in Fig. 10a). If we compare cases with similar ELAs, which imply similar
glacier sizes, our results indeed show an increasing trend of glacial erosion rates as mean annual temperatures rise (Figs. 4 and
10), although the amount of increase in glacial erosion predicted by our model (10-fold) is less than the over 100-fold difference
observed by Koppes et al. (2015). Importantly, our results support the idea of Koppes et al. (2015) that glacial erosion is highly
variable in a relatively narrow range of climates as a result of changes in basal thermal regime.

The temporal evolution of glacial erosion rates inferred from sedimentary records suggests that the response of glacial
erosion to climate forcing is nonlinear and that glacial erosion preferentially occurs during short periods with optimal climatic
conditions (Fernandez et al., 2011; Ganti et al., 2016; Mariotti et al., 2021). Mariotti et al. (2021) suggest that such nonlinear
forcing of climate is a result of the complex interplay between glacier sliding velocity and topography. In this study, our

simulations predict a wide range of glacial erosion rates due to the climatically controlled basal thermal regime and a cold and wet climate is the optimal condition for rapid glacial erosion. This finding provides an alternative mechanism for the nonlinear relationship between glacial erosion and climate. The highly variable erosion rates also provide implications for the ongoing debate on the potential global increase in erosion rates in response to widespread glaciations during the Pleistocene (Herman et al., 2013; Herman and Champagnac, 2016; Willenbring and Jerolmack, 2016). Our results suggest that, due to the variation of basal thermal regimes in different climatic settings, glaciations in cold and dry regions do not necessarily induce rapid glacial erosion.

## 5 Conclusions

In this study, we investigate the impact of climatic conditions on the basal thermal regime of glaciers and glacial erosion patterns, using a landscape evolution model coupled with an ice sheet model. Our results indicate that the spatial patterns of glacial erosion follow the patterns of the basal thermal regime. Cold temperatures create cold-based glacier areas at high elevations, while high precipitation rates tend to cause warm-based conditions by increasing the thickness of glaciers and lowering the melting point of ice. Glaciers in a cold and dry climate have limited erosion at high elevations due to cold-based conditions, and most glacial erosion focuses at low elevations in major valleys. By contrast, a warm and wet climate causes a large amount of erosion at high elevations. Our results do not support the direct correlation between the ELA and the patterns of glacial erosion, because different temperature and precipitation combinations could produce glaciers with similar ELAs but distinct basal thermal regimes. Our study provides a mechanistic basis for the relationship between climate and glacial erosion, and reinforces the interactions between climate and erosional processes.

**Code availability**

The version of PISM used in this study is available at https://github.com/laijingtao/pism/tree/jlai-Esurf-2021.

**Author contribution**

JL and AA designed the experiments, and JL developed code and conducted the experiments. Both authors contributed to data analysis. JL prepared the manuscript with inputs from AA.

**Competing interests**

The authors declare that they have no known conflict of interest.

**Acknowledgments**

We are grateful to Andy Aschwanden and Jonathan Tomkin for their constructive comments on multiple drafts of the
manuscript. We thank Simon Cook and Ian Delaney for their constructive review.

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

## Figures and tables

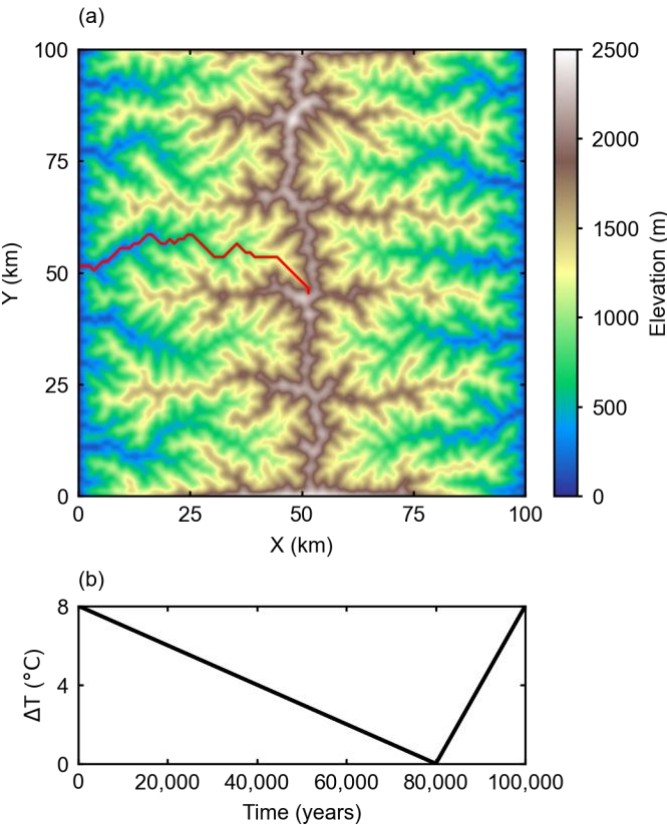

Figure 1: (a) The initial bedrock topography is a synthetic fluvial landscape, representing a typical pre-glacial setting. The piedmont plains are not shown in the figure for a clear illustration. The red curve indicates the valley profile shown in Fig. 4. (b) Cyclic climate forcing. The mean annual sea-level temperature decreases linearly for 80,000 years and then rises for 20,000 years. The magnitude of the temperature change is 8 °C. All simulations use the same cyclic climate forcing and the climatic conditions at glacial maximum (80,000 years) are different in different cases.

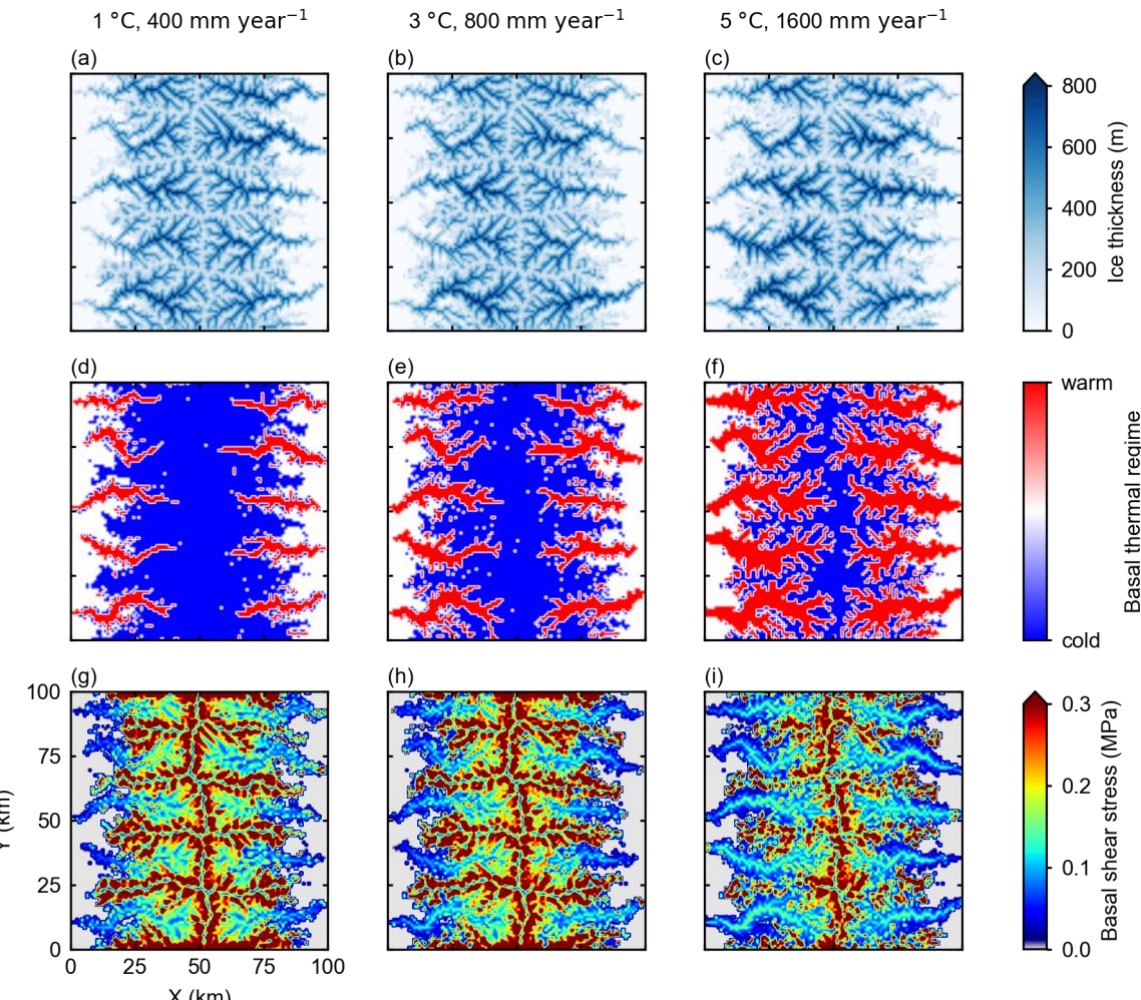


**Figure 2: 2-D mapview of modeled ice thickness (a-c), basal thermal regime (d-f), and basal shear stress (g-i) at glacial maximum. The left column (a and d) is the case with a mean annual sea-level temperature of 1 °C and a mean annual precipitation rate of 400 mm year[-1] at the glacial maximum, corresponding to a glacial ELA of 1300m. The middle column (b and e, glacial mean annual sea-level temperature = 3 °C, glacial mean annual precipitation = 800 mm year[-1]) and right column (c and f, glacial mean annual sea-level temperature = 5 °C, glacial mean annual precipitation = 1600 mm year[-1]) are cases with warmer and wetter climate than the**
**left column. These three climatic settings produce similar ELAs around 1300 m at glacial maximum.**

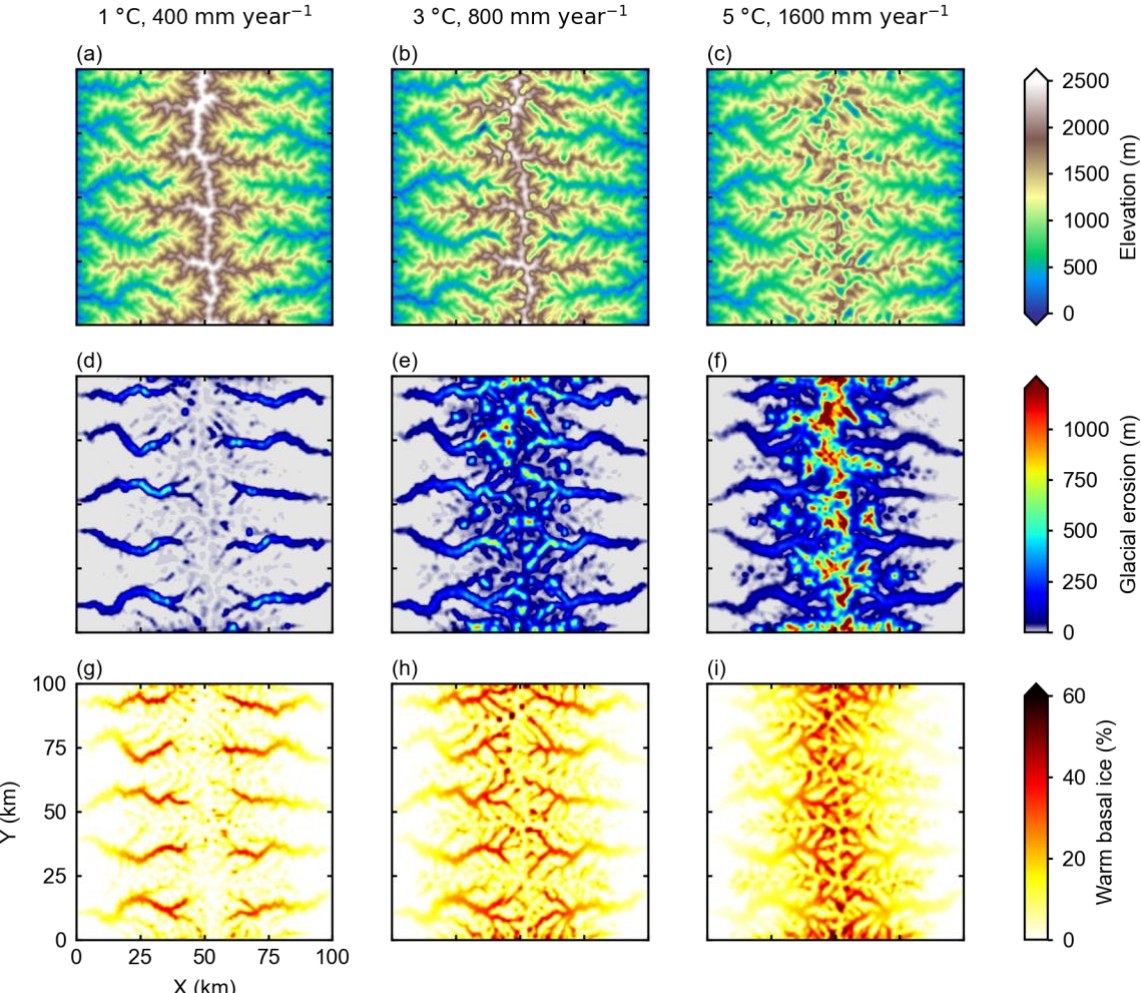

**Figure 3: 2-D mapview of the modeled topography after a glacial-interglacial cycle (a-c), amount of glacial erosion (d-f) and percentage of time with warm basal ice (g-i). Each column represents model results for a specific climate. The three climatic settings produce similar ELAs around 1300 m.**

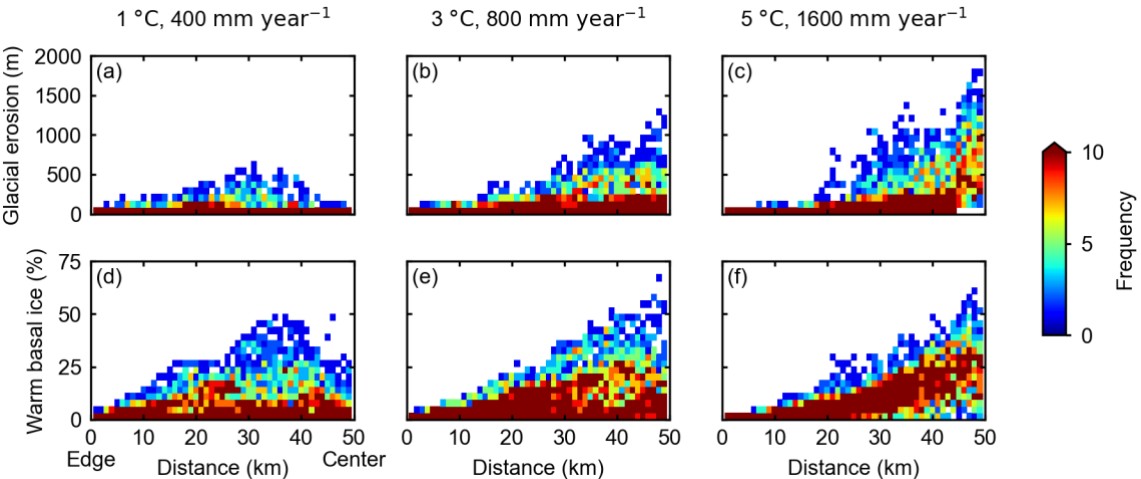

**Figure 4: Spatial variability in glacial erosion (a-c) and percentage of time with warm basal ice (d-f). The x-axes are the distance from the left or right edge of the domain. The color scheme represents the frequency of pixels for a given combination of glacial erosion/percentage of time with warm basal ice and distance. Each column represents model results for a specific climate. The three climatic settings produce similar ELAs around 1300 m.**


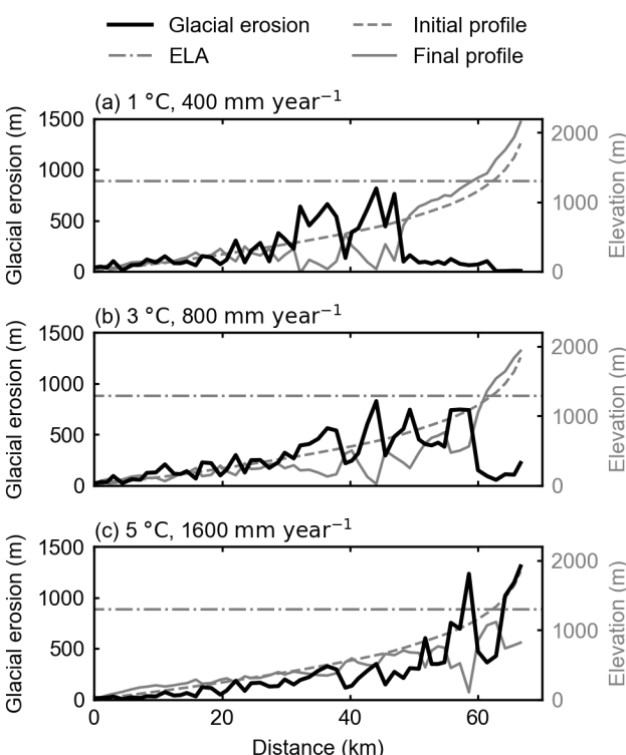

**Figure 5: Glacial erosion (black lines), initial elevation (gray dashed lines) and finial elevation (gray solid lines) along a valley long profile. The location of the valley profile is shown as a red curve in Fig. 1a. Horizontal gray dash-dotted lines represent glacial ELAs. Although the glacial ELAs are similar in three cases, the spatial patterns of glacial erosion are different.**

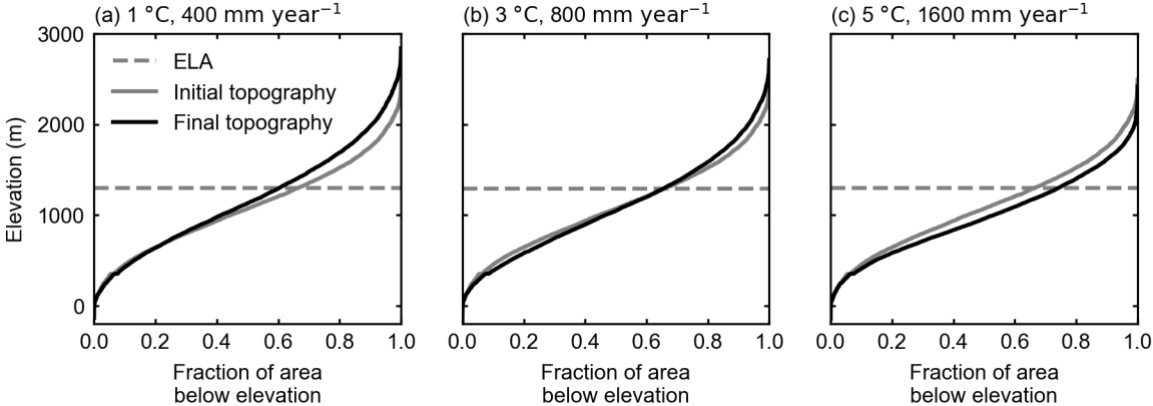

**Figure 6: Hypsometric evolution of modeled landscapes in different climates. Initial topography is shown in gray solid lines and final topography is shown in black. Horizontal gray dashed lines indicate the ELAs.**


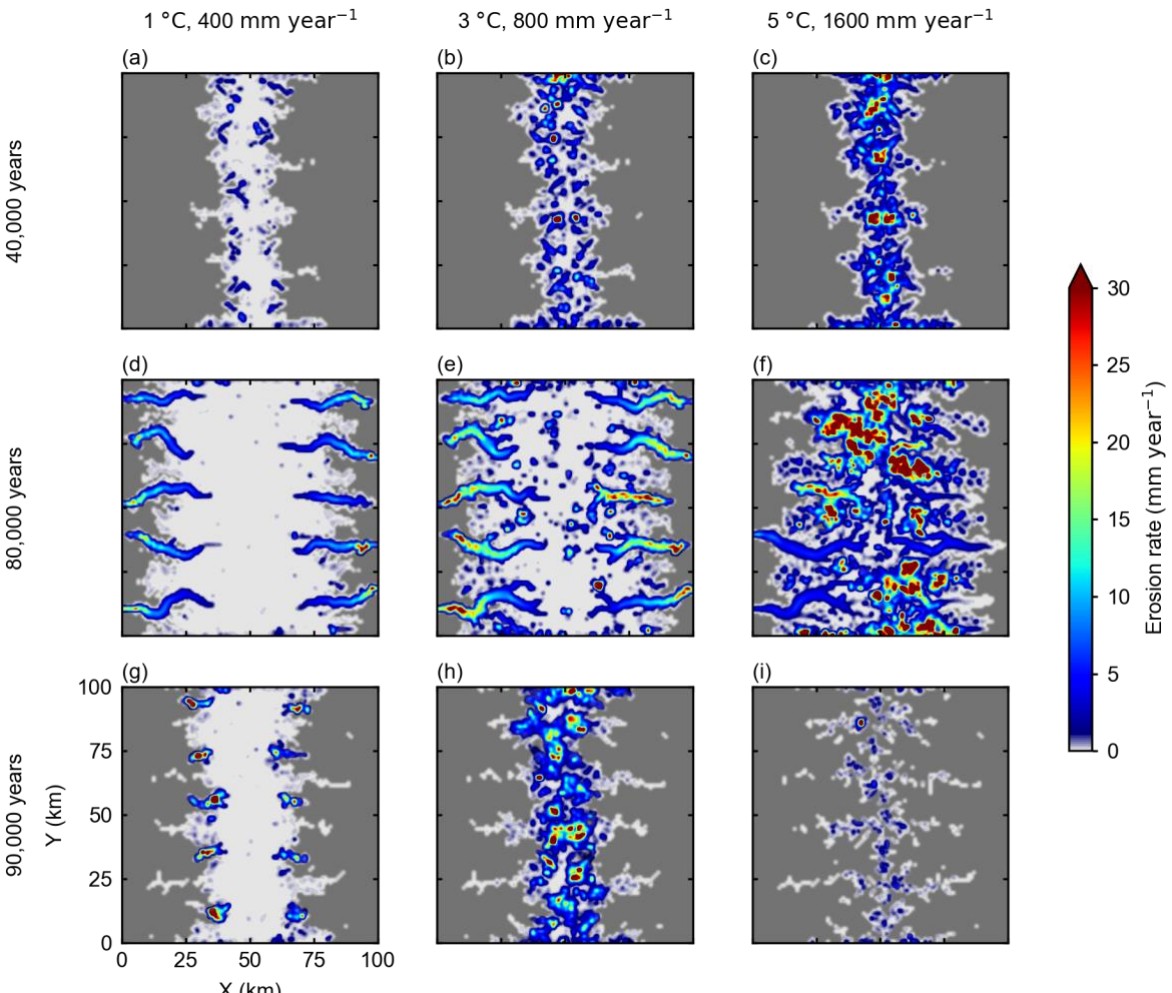

**Figure 7: Average glacial erosion rates during three 1,000-year windows (40,000, 80,000, and 90,000 years after the simulation starts) for three cases with different climatic settings that produce similar ELAs around 1300 m. Each column represents simulation results for a specific climate and each row represent a time window. Ice-free areas are shown as dark gray.**

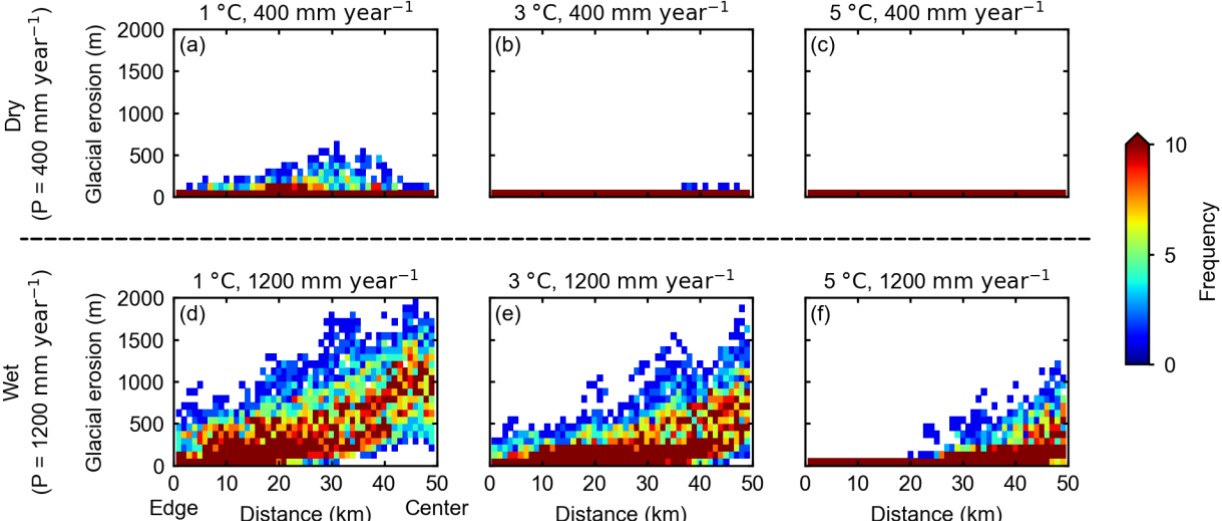

**Figure 8: Influence of temperature on the spatial variability in glacial erosion. Each panel represents the result for a specific climate. (a-c): The mean annual precipitation rate is 400 mm year-1 at glacial maximum in all three cases and the mean annual sea-level temperatures are 1, 3, and 5 °C, respectively. (d-e): The mean annual precipitation rate is 1200 mm year-1 at glacial maximum in all three cases and the mean annual sea-level temperatures are 1, 3, and 5 °C, respectively.**


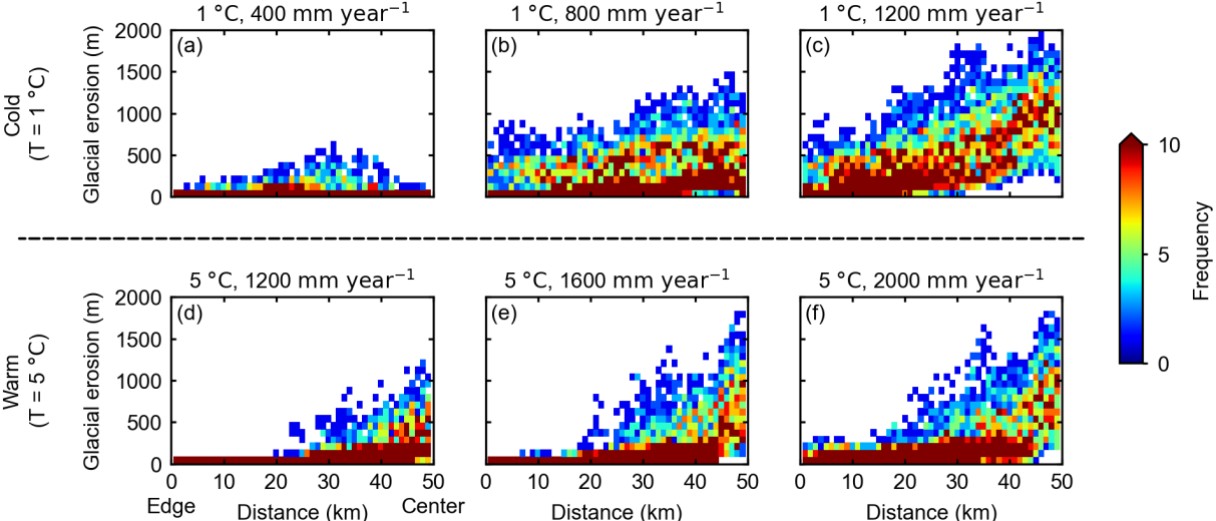

**Figure 9: Influence of precipitation on the spatial variability in glacial erosion. Each panel represents the result for a specific climate. (a-c): The mean annual sea-level temperature is 1 °C at glacial maximum in all three cases and the mean annual precipitation rates at glacial maximum are 400, 800, and 1200 mm year-1, respectively. (d-e): The mean annual sea-level temperature is 5 °C at glacial**
**maximum in all three cases and the mean annual precipitation rates at glacial maximum are 1200, 1600, and 2000 mm year-1, respectively.**

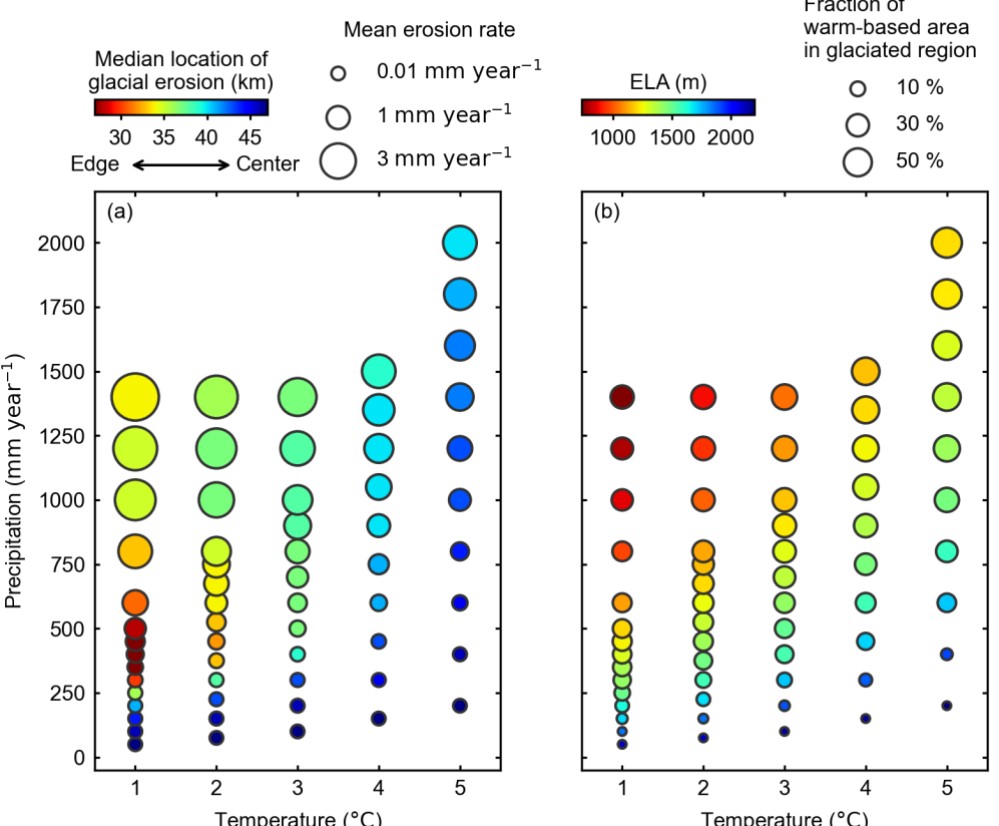

**Figure 10: (a)** The rates and median locations of glacial erosion in different climatic settings. Horizontal and vertical axes indicate the mean annual sea-level temperature and mean precipitation during the glacial maximum, respectively. The size of the circle represents the mean erosion rates in glaciated regions. The color scheme indicates the median locations of glacial erosion (see definition in Section 3.5). Red colors mean most erosion occurs near the edge of the mountain range, and blue colors represent that glacial erosion focuses near the center of the mountain range. **(b)** The ELAs and fraction of warm-based areas in glaciation-only cases in different climatic settings. Similarly, horizontal and vertical axes indicate climatic conditions. The size of the circle shows the fraction of warm-based area in glaciated regions during the glacial maximum, and the color scheme represents the ELAs during the glacial maximum.

Table 1 Climate conditions explored in this study

| Mean annual sea-level temperature at glacial maximum (°C) | Mean annual precipitation at glacial maximum (mm year$^{-1}$) |
|---|---|
| 1 | 50, 100, 150, 200, 250, 300, 350, 400, 450, 500, 600, 800, 1000, 1200, 1400 |
| 2 | 75, 150, 225, 300, 375, 450, 525, 600, 675, 750, 800, 100, 1200, 1400 |
| 3 | 100, 200, 300, 400, 500, 600, 700, 800, 900, 1000, 1200, 1400 |
| 4 | 150, 300, 450, 600, 750, 900, 1050, 1200, 1350, 1500 |
| 5 | 200, 400, 600, 800, 1000, 1200, 1400, 1600, 1800, 2000 |
