# Peer review of "Climatic controls on mountain glacier basal thermal regimes dictate spatial patterns of glacial erosion"

_Earth Surface Dynamics, 2021_

## Referee Comment (RC2)

**Review of "Climatic controls on mountain glacier basal thermal regimes dictate spatial patterns of glacial erosion"**

**General comments**

Dear Editor,

Thank you for the opportunity to review this paper. This paper discusses the role of a glacier's thermal regime on glacier erosion. To accomplished this, the authors use PISM, a well-established model for the thermo-mechanical physics of ice flow. The authors argue that in warmer climates, a greater proportion of the glacier bed can experience sliding and thus erosion is increased in a warmer climate. Additionally, greater erosion can occur in warm/wet conditions, with a high mass-balance gradient, than on a cold/dry glacier. Thus, a glacier's ELA does not greatly affect it's capacity to erosion a landscape.

I like this paper. It is well written, concise and organized. Generally, the experiments and conclusions are thoughtfully executed. In my opinion, much work remains to evaluate the interaction between climate and glacier erosion, and I applaud the authors for pursuing this topic. This is highly important given that increased knowledge of glacial erosion processes will help to interpret the sedimentary records of erosion, understand the landscape response to the forthcoming deglaciation and better evaluate the relationship between tectonic uplift and glacier erosion. Implementing a complex model that considers these processes cohesively definitely contributes to our knowledge. In its most basic sense, these results show the complexity of glacial erosion. In this paper, however, I have some questions about the conclusions drawn from these numerical experiments and the experiment design. For me, personally, to subscribe to the conclusions of this paper, they will need to be answered fully.

1. Most importantly, to what degree is basal shear stress (largely influenced by driving stress), and not basal temperature, responsible for the change in glacier sliding, and thus erosion? Lower glacier surface gradients at high elevations may cause a decrease in sliding and thus erosion there. This is discussed in the Methods of the paper where the Weertman-style sliding law is explained. Changes in driving stress are what Seguinot and Delaney (esurf-dis) suggest as the process responsible for variations in erosion. Given the experiment design, basal thermal conditions might be a factor, however, the role and effects of basal shear-stress (and other impacts on glacier sliding) should be thoroughly considered and addressed.

   Some changes in erosion might be linked to basal conditions. However, the role of increased precipitation in glacier erosion suggests to me that glacier topography and in morphology could well play a role in the increased erosion observed in those model runs. At high elevations, increased precipitation means increased ice flux down glacier and potentially high driving stresses.

   For me, these processes (basal temperature and shear stress) should be attributed clearly.

2. The time transgressive nature of the climate forcing is not addressed. I was left wondering why this forcing was applied, as opposed to a steady forcing, given the lack of discussion about temporal variations in erosion. By examining temporal patterns of glacier erosion, the authors may find that similar basal conditions produce different erosion rates due to changes in glacier morphology and driving stress. If this is the case, the authors might be able argue that basal conditions are responsible for background erosion rates, however, variations in erosion largely come from changes in climate and glacier dynamics. Nonetheless, these timescales and processes should be parsed and explained in order to present the cases where basal temperature is the primary control on glacier erosion. In this way, the findings herein and from Seguinot and Delaney (esurf-dis) might be able to complement each other.

3. Related to the last question, the response time of the glacier temperature changes is very slow and it can take a long time for the glacier to reach an equilibrium. I did not adequately understand the initial conditions prescribed to the glacier and I am uncertain the degree to which these conditions could propagate into the results. Basically, it seems that the conclusion that basal temperature drives the erosion rate might be a manifestation of the initial condition chosen and not necessarily the time transgressive response to variations in temperature. The impact of the initial condition could well be a reasonable result, however, it should be made clear and it is a limited component of how the natural system operates and responds to climate. Some commentary on the response of glacier sliding and temperature to climate, and the time scale there of, will help with some of these results.

A revised manuscript should address these comments above. More specific comments are given below.

Hopefully the authors find my comments well-guided. As stated above, I believe this to be an important topic to pursue, and I am glad to see papers being pursued to these ends. I wish the authors the best in concluding and presenting this research and look forward to seeing the final result.

Best regards

Ian Delaney

**Specific Comments**

- **Introduction** I found the introduction well written. A couple of notes though. 1) I would recommend discussing glacier sliding more and the contributing factors. The effective manifestation of erosion here is sliding, so sliding is of interest. Also 2) consider discussing the role of understanding the glacier dynamics in the context of interpreting the sedimentary record (papers by Koppes, Fernandez and Ganti). Also, I found many unattributed statements in the introduction. Find citations for these or omit.

- **Ln 66–79** The findings of Anderson et al. (2012) may well fit well in to this paragraph.

- **Methods** In addition to the comments in the letter, some issues arose in this section. What are the initial conditions of the glacier model? Is a spin up established? Is the fluvial topography in a steady state? If not, how long was the model run for prior to the initialization of the model run? Are the fluvial and glacier models running concurrently over the same parts of the domain during the model runs?

- **Section 2.2.1** A linear erosion rule is used, which should be attributed to Humphrey and Raymond (1994). However, Herman et al., (2015), Koppes et al. (2015), and Cook et al. (2020) all empirically explain why a non-linear law likely fits better. Furthermore, the theory in Hallet (1979) suggests that an exponent of 2 should be used. Herman et al. (2021) discusses this in detail. I am not saying this erosion rule is wrong, but it should be justified. Also, work by Humphrey and Raymond (1994), Herman et al., (2015), Koppes et al. (2015), and Cook et al. (2020) all implement data to validate the rule, so citing them not only provides proper attribution, but strengthens the method.

- **Ln 134** "Fluvial landscape in Landlab"... what is the relationship between this and the Braun and Willet 2013 paper discussed above. Also Deal and Prasicek (2020) might be able to provide some good insight in to fluvial glacial relationships, depending on the relationship between fluvial and glacial erosion.

- **Section 2.4/2.5** I found some of these aspects of the design complicated. Would a cartoon or timeline with climate forcing and model interaction in addition to the map in Fig. 1 help clarify?

- **Ln 165** "PISM over an unchanging topography." I am a bit confused by this statement as I though topography was evolving (line 100).

- **Results** As stated in the letter, basal conditions are not the only driver of glacier sliding, so basal shear-stress should be implemented into this results and findings.

- **Ln 214** I was expecting a section titled "Temporal patters of glacier erosion" here.

- **Section 3.3** As mentioned the letter, the amount of time needed for the englacial temperature to adjust to the climate can be very large (10's of thousands of years). It seems that some comments on the basal temporal variations and the response time of the glacier bed to atmospheric forcing would help this section.

- **Section 3.4** Coupled with sensitivity to temperature, precipitation can affect glacier morphology and thus the stress balance. Generally, glaciers with a steeper mass balance gradient will slide faster and thus may increase in erosion.

- **Section 3.5** Nice commentary. I am glad this is discussed.

- **Ln 294–305** The findings of Anderson et al. (2012) could fit well here. Also, it may be appropriate to find an alternate explanation (average glacier conditions), but do findings here support that?

- **Ln 303–305** Here some temporal evolution topics are discussed, more would be appreciated.

- **Ln 294–312** A recent paper by Mariotti et al. (2021) high-lights the impact of climate on glacial erosion in the sediment record. I believe this paper could help the authors and give context to some findings.

- **Ln 315** "role of precipitation. . . ." Cook et al. 2020 speaks to this.

- **Ln 317** What process is responsible for precipitation changing the thermal regime? This needs some explanation, and does the relevant process fit the timescale of the erosion in the model run?

- **Ln 329–333** This is true and precipitation is not always represented well. However, it can be represented by having a large (steep) mass balance gradient in a linear mass-balance forcing relationship. Some caution should used here.

- **Ln 334–339** Comparison with Koppes et al. (2015) is very difficult. Some of the differences are discussed (i.e. land vs marine terminating). However, such a comments also requires considering the evolution of glaciers as they respond to climate.

**Figures**

- **Figs 1–3** There is a black box around the figs in the print version. Not a big deal but worth a thought.

- **Figs 4** Would it wise to plot "Warm basal ice" vs "Glacial erosion"?

---

## Author Comment (AC1)

Dear Dr. Cook,

Thank you for your comments on our manuscript. We appreciate your constructive and thoughtful review. The line-by-line comments are very helpful and will be incorporated into the revised manuscript. I'll respond to the major comments below, with the reviewer's comment shown as italic and our response as normal font.

*L46 – I'm not sure you can quite say that glacial erosion varies as a function of basal thermal regime because Koppes et al. (2015) used mean annual air temperature rather than basal temperature.*

We agree. Glacial erosion varies as a function of the basal thermal regime is not precisely the conclusion of Koppes et al. (2015), but their results do imply this point. We will reword this sentence in the revised manuscript.

*L66 – I didn't really understand this sentence. It starts off being about polythermal glaciers, but ends in making the same point you have made several times already about thermal regime needing more study.*

Sorry for the confusion. The point of this sentence is that previous work on the impact of thermal regimes on glacial erosion mainly focuses on comparing the thermal regimes between different glaciers, rather than comparing the different portions within a polythermal glacier. For example, people have been classified glaciers as cold vs warm glaciers and have suggested that cold glaciers could protect mountains while warm glaciers destruct mountains. However, polythermal glaciers are more common than purely cold or warm glaciers. Therefore, we need to pay more attention to the variations of thermal regimes within a glacier/icecap. We will rephrase this sentence in the revised version.

*L105-117 – This section discusses the glacial erosion rule employed in the modelling effort; the authors use a linear erosion rule and justify that choice with reference to previous studies that also assume erosion rate to be a linear function of sliding velocity. Nonetheless, several papers have been published since those cited here that suggest that the sliding velocity be raised to some exponent (l) which could be <1 (Cook et al., 2020), ~2 (Herman et al., 2015), or >2 (between ~2.3-2.6; Koppes et al., 2015). I wonder if this should be mentioned in the*

*manuscript. I don't think there is any problem with the approach used by the authors, but the justification of the erosion rule used seems one-sided. Cook et al. (2020) suggested that an exponent of 2 would be suitable for single glaciers and an exponent of 1 or less would be appropriate for ice caps/sheets & mountain ranges comprising multiple glaciers – so their work potentially supports your choice of erosion rule formulation.*

Thanks for the comment. This is a good point. We chose this value because our aim is to investigate the spatial pattern of glacial erosion and different values of the exponent have little impact on this spatial pattern. We will provide more references for different choices for the value of the exponent in the revised manuscript.

*L123 – do you need to justify (e.g. using citations) why you have selected these values for the constants in the fluvial incision model? There was a lot of justification for the use of a linear glacial erosion rule, but the same detail is not here for the fluvial model.*

Thanks for the suggestion and we do need to justify this. We will add a short discussion about our choice of model and model parameters in the revised manuscript.

*L327-8 – it's probably a bit self-centred to suggest it, but Cook et al. (2020) provided direct empirical evidence from modern erosion rates and precipitation rates for the influence of precipitation on erosion. We even found that precipitation explained more of the variability in the erosion rate data than did temperature. I wonder if this could/should be mentioned here in your Discussion – certainly, it supports the point you are making here.*

Thanks for the suggestion. We did read your work but we must have missed this point you mentioned. We will add this in the revised manuscript.

*Fig 9 – I might be misinterpreting (or over-interpreting?) this diagram, but it seems to me that there is a systematic increase in erosion rate with increasing precipitation; there is not the same systematic increase in erosion rate with increasing temperature. We (Cook et al., 2020) found the same relationship (our Figure 3b and 3c). Perhaps this provides empirical support for your results.*

Your interpretation is correct. However, it is important to notice that the climate range we explored here is much smaller than your work, especially the temperature range. This is why we did not make any direct comparison with your Fig 3 (and Fig 4) in this manuscript. We've cited your work in the discussion section and we decided not to oversell our results in Fig 9.

Best,

Jingtao Lai

---

## Author Comment (AC2)

Dear Dr. Delaney,

Thank you for your review of our manuscript. We appreciate your thoughtful review and constructive comments. We will respond to the comments below, with the reviewer's comments shown in italic and our response in normal font.

*1. Most importantly, to what degree is basal shear stress (largely influenced by driving stress), and not basal temperature, responsible for the change in glacier sliding, and thus erosion? Lower glacier surface gradients at high elevations may cause a decrease in sliding and thus erosion there. This is discussed in the Methods of the paper where the Weertman-style sliding law is explained. Changes in driving stress are what Seguinot and Delaney (esurf-dis) suggest as the process responsible for variations in erosion. Given the experiment design, basal thermal conditions might be a factor, however, the role and effects of basal shear-stress (and other impacts on glacier sliding) should be thoroughly considered and addressed.*
*Some changes in erosion might be linked to basal conditions. However, the role of increased precipitation in glacier erosion suggests to me that glacier topography and in morphology could well play a role in the increased erosion observed in those model runs. At high elevations, increased precipitation means increased ice flux down glacier and potentially high driving stresses.*
*For me, these processes (basal temperature and shear stress) should be attributed clearly.*

Thanks for the comment. We agree that basal shear stress is another important factor that can potentially influence basal sliding in our simulations, in addition to the basal thermal regime. PISM can calculate the basal shear stress and we plot the results in Fig. R1a-c. The warm and wet case (Fig. R1c) predicts slightly higher basal shear stress in major valleys than the cold and dry case (Fig. R1a). This can explain the faster erosion rates in the warm and wet case than the cold and dry case. However, the basal shear stress does not show a significant variation in spatial patterns between these cases, especially in major valleys. This makes our point stronger that climate influences the spatial patterns of glacial erosion through the basal thermal regime.

We also want to point out that the differences in basal shear stress are not caused by differences in driving stress. We calculate the driving stress based on ice thickness and ice surface gradient (Fig. R1d-f). The results show that the driving stress distributions are very

similar in the three cases, especially in the major valleys where we observe different basal shear stress (Fig. R1d-f). This is because these climate settings produce glaciers with similar ELAs, therefore, similar ice thickness, which eventually gives us similar driving stresses. The similar driving stress patterns are implied by our Fig. 2, where we show that the ice thickness distributions are similar in the three cases. We interpret the differences in basal shear stress as a result of the temperature-dependent ice stiffness. In a cold climate, the ice is stiffer and therefore a higher fraction of the driving stress is balanced by the membrane stress within the ice than in a warm climate. Thus the basal shear stress is lower in a cold climate.

We will add a discussion about basal shear stress in the revised manuscript. Thanks for the constructive comment.

[Figure]

Figure R1 Driving stresses (a-c) and basal shear stresses (d-f) in three cases with similar ELAs.

*2.The time transgressive nature of the climate forcing is not addressed. I was left wondering why this forcing was applied, as opposed to a steady forcing, given the lack of discussion about temporal variations in erosion. By examining temporal patterns of glacier erosion, the authors may find that similar basal conditions produce different erosion rates due to changes in glacier morphology and driving stress. If this is the case, the authors might be able argue that basal conditions are responsible for background erosion rates, however, variations in erosion largely come from changes in climate and glacier dynamics. Nonetheless, these timescales and processes should be parsed and explained in order to present the cases where*

*basal temperature is the primary control on glacier erosion. In this way, the findings herein and from Seguinot and Delaney (esurf-dis) might be able to complement each other.*

Thanks for your comment. In our earlier work (Figs. 4 and 5 in Lai and Anders, 2020, EPSL) we found that a cyclic climate produces more erosion at high elevations than a steady climate, simply because high-elevation regions are covered by glaciers for a longer time than low elevations in a glacial-interglacial cycle. Therefore, we used a cyclic forcing in this study to get more "realistic" patterns of glacial erosion.

You raised a good point on temporal variations in erosion caused by changes in glacial morphology and driving stress. We did examine the temporal changes in glacial erosion in our experiments. In the figure below, we plot the glacial erosion rate during a 1000-year window at 40,000, 80,000, and 90,000 years. The climate reaches the coldest glacial period at 80,000 years, and the climatic conditions are the same at 40,000 and 90,000 years. We observe an inward migration and an acceleration of glacial erosion during deglaciation in the cold and dry case (Fig. R2g), which is similar to your findings in Seguinot and Delaney (Esurf diss.). Your simulations suggest that such a change in erosion is caused by the change in ice surface gradient, which influences the basal shear stress. In our simulations, this mechanism may contribute to the change in erosion, but we cannot exclude the possibility that the feedbacks between eroded topography and glacier dynamics may also lead to an increase in basal shear stress. The deeper valleys and steeper slopes in the eroded topography allow for thicker glaciers with a steeper surface gradient, which presumably can raise the basal shear stress.

We will add a paragraph describing the temporal evolution of glacial erosion in the revised manuscript.

[Figure]

Figure R2 Temporal evolution of glacial erosion rates.

*3.Related to the last question, the response time of the glacier temperature changes is very slow and it can take a long time for the glacier to reach an equilibrium. I did not adequately understand the initial conditions prescribed to the glacier and I am uncertain the degree to which these conditions could propagate into the results. Basically, it seems that the conclusion that basal temperature drives the erosion rate might be a manifestation of the initial condition chosen and not necessarily the time transgressive response to variations in temperature. The impact of the initial condition could well be a reasonable result, however, it should be made clear and it is a limited component of how the natural system operates and responds to climate. Some commentary on the response of glacier sliding and temperature to climate, and the time scale there of, will help with some of these results.*

Thanks for the comment, it's a good point. We agree that the response time of glacier temperature change is important and therefore, the initial conditions can influence the results.

For this reason, our experiment did not aim to compare the temporal variations in basal thermal regimes in response to climate change during a glacial-interglacial cycle. What we compare in our experiments is the basal thermal regimes in different "regions" with different climates. For example, in Figs. 2 and 3, "1 ℃, 400 mm/year" means that during the coldest glacial period the mean annual sea-level temperature is 1 ℃ and the mean annual precipitation is 400 mm/year. Similarly, the "5 ℃, 1600 mm/year" case has a warmer and wetter climate. All the simulations have a temperature change of 8 ℃ from the warmest interglacial time to the coldest glacial period, and the initial topography is ice-free in all cases. The ice-free initial condition is reasonable given that the 8 ℃ temperature change only allows for a very limited ice cover during the interglacial periods in most cases. When we compare the results of "1 ℃, 400 mm/year" and "5 ℃, 1600 mm/year", we are actually comparing two mountain ranges at two "locations" with different climates, rather than investigating the temporal variation of basal thermal regimes in a mountain range with climate changing from a cold and dry one to a warm and wet one.

*Specific Comments*
*• **Introduction** I found the introduction well written. A couple of notes though. 1) I would recommend discussing glacier sliding more and the contributing factors. The effective manifestation of erosion here is sliding, so sliding is of interest.*

Thanks for the suggestion. The main purpose of this study is to examine how climate changes the basal thermal conditions and consequently, basal sliding/glacial erosion. Therefore, in the introduction, we try to make a point that the basal thermal regime is a first-order control on where erosion can occur. Furthermore, although our model uses a sliding-based erosion law, that does not mean basal thermal regime cannot influence erosion in other ways. Therefore, we focus on the discussion of how the basal thermal regime may control glacial erosion in different ways. In summary, while basal shear stress, subglacial hydrology, etc. are important factors that control the rate of sliding, we think it is unnecessary and probably misleading to discuss all of them.

*Also 2) consider discussing the role of understanding the glacier dynamics in the context of interpreting the sedimentary record (papers by Koppes, Fernandez and Ganti).*

This is a very nice suggestion. We will add this in the introduction and also in the discussion.

*Also, I found many unattributed statements in the introduction. Find citations for these or omit.*

Most parts of the introduction have citations to support them. We assume you are referring line 57-65 because we think this the only part with some unattributed statements. We are also unsatisfied with this part because this is a description of the influence of climate on basal thermal regimes in the model rather than in real glaciers. Therefore, we will remove this part in the revised manuscript.

• **Ln 66–79** *The findings of Anderson et al. (2012) may well fit well in to this paragraph.*

Thanks for the suggestion. We will add a citation to Anderson et al. (2012).

• **Methods** *In addition to the comments in the letter, some issues arose in this section. What are the initial conditions of the glacier model? Is a spin up established? Is the fluvial topography in a steady state? If not, how long was the model run for prior to the initialization of the model run? Are the fluvial and glacier models running concurrently over the same parts of the domain during the model runs?*

Most of these issues have already been described in the original manuscript. We will add a description of the initial conditions of the glacier dynamics model in the revised manuscript.

• **Section 2.2.1** *A linear erosion rule is used, which should be attributed to Humphrey and Raymond (1994). However, Herman et al., (2015), Koppes et al. (2015), and Cook et al. (2020) all empirically explain why a non-linear law likely fits better. Furthermore, the theory in Hallet (1979) suggests that an exponent of 2 should be used. Herman et al. (2021) discusses this in detail. I am not saying this erosion rule is wrong, but it should be justified. Also, work by Humphrey and Raymond (1994), Herman et al., (2015), Koppes et al. (2015), and Cook et al. (2020) all implement data to validate the rule, so citing them not only provides proper attribution, but strengthens the method.*

We will add a short discussion to justify this. Thanks for the suggestion.

• *Ln 134* *"Fluvial landscape in Landlab"...what is the relationship between this and the Braun and Willet 2013 paper discussed above. Also Deal and Prasicek (2020) might be able to provide some good insight in to fluvial glacial relationships, depending on the relationship between fluvial and glacial erosion.*

Landlab implements Braun and Willet 2013 paper's method for node ordering, which provides an efficient way to calculate drainage area. Deal and Prasicek (2020) is a valuable contribution, but we don't think this is very relevant since we do not aim to compare the steady-state glacial landscapes with fluvial landscapes in our work.

• *Section 2.4/2.5* *I found some of these aspects of the design complicated. Would a cartoon or timeline with climate forcing and model interaction in addition to the map in Fig. 1 help clarify?*

We will add a figure of the glacial/interglacial forcing in Fig. 1.

• *Ln 165* *"PISM over an unchanging topography." I am a bit confused by this statement as I though topography was evolving (line 100).*

We run two simulations for each climate, one with erosion and one without erosion.

• *Results* *As stated in the letter, basal conditions are not the only driver of glacier sliding, so basal shear-stress should be implemented into this results and findings.*

We will add a discussion about basal shear stress in the revised manuscript.

• *Ln 214* *I was expecting a section titled "Temporal patters of glacier erosion" here.*

We will add a paragraph talking about the temporal patterns.

• *Section 3.3* *As mentioned the letter, the amount of time needed for the englacial temperature to adjust to the climate can be very large (10's of thousands of years). It seems that some comments on the basal temporal variations and the response time of the glacier bed to atmospheric forcing would help this section.*

As we explained above, it is not our goal to evaluate the temporal evolution of the basal thermal regime in response to climate change during a glacial-interglacial cycle.

• *Section 3.4 Coupled with sensitivity to temperature, precipitation can affect glacier morphology and thus the stress balance. Generally, glaciers with a steeper mass balance gradient will slide faster and thus may increase in erosion.*

We agree. In this section we focus on the spatial patterns of glacial erosion and our point is that precipitation changes the basal thermal regime and therefore the patterns of erosion.

• *Section 3.5 Nice commentary. I am glad this is discussed.*

Thanks!

• *Ln 294–305 The findings of Anderson et al. (2012) could fit well here. Also, it may be appropriate to find an alternate explanation (average glacier conditions), but do findings here support that?*

A citation to Anderson et al. 2012 will be added.
The alternative explanation is not a conclusion of our results and we do provide references for this argument in the original manuscript. We will add more references to make it clear. Combining this argument and our results suggest that cirque-based ELAs estimation is not a good proxy for glacial erosion patterns.

• *Ln 303–305 Here some temporal evolution topics are discussed, more would be appreciated.*

See response above.

• *Ln 294–312 A recent paper by Mariotti et al. (2021) high-lights the impact of climate on glacial erosion in the sediment record. I believe this paper could help the authors and give context to some findings.*

We will discuss the implication of our results for the long-term temporal evolution of glacial erosion rates in the revised manuscript. Thanks for the suggestion.

• **Ln 315** *"role of precipitation. . . ." Cook et al. 2020 speaks to this.*

This is true, but we think it's still "limited" in comparison with the large amount of literatures looking at the role of precipitation for fluvial and hillslope processes. We cite Cook et al. 2020 later in this paragraph.

• **Ln 317** *What process is responsible for precipitation changing the thermal regime? This needs some explanation, and does the relevant process fit the timescale of the erosion in the model run?*

This is a good point. On Ln 323 in the original manuscript, we interpret this as the changes of the melting point of ice in response to changes in ice thickness due to precipitation. Intuitively, high precipitation rates are able to increase the ice thickness and overburden pressure at the base of ice. Therefore, the melting point of ice is lowered and we can have warm-based basal ice without changing the temperature. We believe this is what happens in the model but it does not necessarily mean it also happens in nature, considering PISM is still a "shallow" model and does not have a vertical flow component. In real glaciers, the vertical flow can also change the englacial temperature field. Because the model is not a perfect representation of nature, we decide not to make an argument on what process is responsible for precipitation changing the basal thermal regime/glacial erosion patterns. Instead, by comparing with field measurements of glacial erosion rates (Cook et al, 2020; Koppes et al. 2015), we suggest that our model can capture the bulk influence of precipitation on glacial erosion, i.e., more precipitation means more erosion.

• **Ln 329–333** *This is true and precipitation is not always represented well. However, it can be represented by having a large (steep) mass balance gradient in a linear mass-balance forcing relationship. Some caution should used here.*

It is true that we could use the mass balance gradient to represent precipitation. Our point here is that the influence of precipitation on the ELA is not adequately represented in this

framework. Usually, the ELA is just set at the elevation with a temperature of 0 degree Celsius.

• **Ln 334–339** *Comparison with Koppes et al. (2015) is very difficult. Some of the differences are discussed (i.e. land vs marine terminating). However, such a comments also requires considering the evolution of glaciers as they respond to climate.*

Thanks for the comment. We will reword this part to make it clear that our results and Koppes et al. are not really an apples-to-apples comparison. In particular, our rates are average glacial erosion rates over a glacial-interglacial cycle, while Koppes et al. report short-term rates that are inferred from sedimentary records.

Figures
• **Figs 1–3** *There is a black box around the figs in the print version. Not a big deal but worth a thought.*

Not sure why. The final manuscript will have figures in pdf formats and hopefully they will be fine.

• **Figs 4** *Would it wise to plot "Warm basal ice" vs "Glacial erosion"?*

We think this is a nice illustration of how the distribution of warm basal ice controls the spatial patterns of glacial erosion.

Best,
Jingtao Lai

References
Lai, J. and Anders, A. M.: Tectonic controls on rates and spatial patterns of glacial erosion through geothermal heat flux, Earth Planet. Sci. Lett., 543, 116348, doi:10.1016/j.epsl.2020.116348, 2020.